# Prevalence of human infection with respiratory adenovirus in China: A systematic review and meta-analysis

**Mei-Chen Liu**[1,2☯]**, Qiang Xu**[2☯]**, Ting-Ting Li**[2,3]**, Tao Wang**[2]**, Bao-Gui Jiang**[2]**, Chen-Long Lv**[2]**, Xiao-Ai Zhang**[2]**, Wei Liu**[1,2]*****, Li-Qun Fang**[1,2,3]*****

**1** Department of Epidemiology and Biostatistics, School of Public Health, Anhui Medical University, Hefei, China, **2** State Key Laboratory of Pathogen and Biosecurity, Beijing Institute of Microbiology and Epidemiology, Beijing, China, **3** Key Laboratory of Environmental Pollution Monitoring and Disease Control, Ministry of Education, School of Public Health, Guizhou Medical University, Guiyang, China

☯ These authors contributed equally to this work.
* lwbime@163.com (WL); fang_lq@163.com (L-Q F)

## Abstract

**Data Availability Statement:** All relevant data are within the manuscript and its Supporting Information files.

### Background

Human adenovirus (HAdV) is a major pathogen that causes acute respiratory tract infections (ARTI) and is frequently associated with outbreaks. The HAdV prevalence and the predominant types responsible for ARTI outbreaks remains obscure in China.

### Methods

A systematic review was performed to retrieve literature that reported outbreaks or etiological surveillance of HAdV among ARTI patients in China from 2009 to 2020. Patient information was extracted from the literature to explore the epidemiological characteristics and clinical manifestations of the infection of various HAdV types. The study is registered with PROSPERO, CRD42022303015.

### Results

A total of 950 articles (91 about outbreaks and 859 about etiological surveillance) meeting the selection criteria were included. Predominant HAdV types from etiological surveillance studies differed from those in outbreak events. Among 859 hospital-based etiological surveillance studies, positive detection rates of HAdV-3 (32.73%) and HAdV-7 (27.48%) were significantly higher than other virus types. While nearly half (45.71%) of outbreaks were caused by HAdV-7 with an overall attack rate of 22.32% among the 70 outbreaks for which the HAdVs were typed by the meta-analysis. Military camp and school were main outbreak settings with significantly different seasonal pattern and attack rate, where HAdV-55 and HAdV-7 were identified as the leading type, respectively. Clinical manifestations mainly depended on the HAdV types and patient's age. HAdV-55 infection tends to develop into pneumonia with poorer prognosis, especially in children <5 years old.

**Funding:** Natural Science Foundation of China (No. 81825019) was awarded to WL. The funders had no role in study design, data collection and analysis, decision to publish, or preparation of the manuscript.

**Competing interests:** The authors have declared that no competing interests exist.

## Conclusions

This study improves the understanding of epidemiological and clinical features of HAdV infections and outbreaks with different virus types, and helps to inform future surveillance and control efforts in different settings.

### Author summary

In this systematic review, we made an exhaustive search of published literature that reported outbreaks or etiological surveillance of HAdV among ARTI patients in China from 2009 to 2020. A total of 950 studies were included in this study, and we explored the epidemiological characteristics and clinical manifestations of the infection of various HAdV types. Positive detection rates of HAdV-3 (32.73%) and HAdV-7 (27.48%) were significantly higher than other virus types according to the hospital-based etiological surveillance studies. Nearly half (45.71%) of outbreaks were caused by HAdV-7 with an overall attack rate of 22.32% among the 70 outbreaks for which the HAdVs were typed by the meta-analysis. Military camp and school were main outbreak settings with significantly different seasonal pattern and attack rate, where HAdV-55 and HAdV-7 were identified as the leading type, respectively. Clinical manifestations mainly depended on the HAdV types and patient's age. HAdV-55 infection tends to develop into pneumonia with poorer prognosis, especially in children <5 years old. This study will help improve the epidemiological and clinical understanding of different HAdV types of human infections and thus will promote the targeted surveillance and measures to control and prevent HAdV infection.

## Introduction

Infection with human adenovirus (HAdV) causes a broad spectrum of clinical illnesses, e.g., pharyngoconjunctival fever, keratoconjunctivitis, pneumonia, hemorrhagic cystitis, gastroenteritis, acute respiratory disease, cardiomyopathy, and encephalitis, which varies depending on the infecting virus types and is more severe among immunocompromised patients such as organ transplant patients [1]. Even unexplained liver injury or hepatitis was reported by two recent independent studies with UK children with HAdV-2 infection, suggesting that HAdV-2 may trigger liver damage through the immune mechanisms of genetically predisposed children [2,3]. There are at least 113 recognized HAdV types (http://hadvwg.gmu.edu/), which are assigned to seven subgroups (A–G) according to biophysical, biochemical, and genetic characteristics, with marked differences in tissue tropism and clinical manifestations [4]. Species C, species B, subspecies B1 and B2 were the most common HAdV types found in respiratory samples among pediatric patients with ARTI [5,6].

In recent years, new serotypes or subspecies were increasingly recognized by using phylogenetic analysis, which arise from genome recombination between the hexon gene, fiber, and penton genes. For any of the emerging new types or recombinant strains, there is a high potential of spreading widely and causing epidemic outbreaks, due to the lack of herd immunity and specific vaccine intervention, posing severe threats to public health [7–10]. Acute respiratory infection caused by HAdV is the leading cause of morbidity in military forces worldwide. Since 1971, U.S. military recruits have been vaccinated with oral HAdV-4 and HAdV-7 vaccines, which has significantly decreased the epidemics of HAdV in the military [1,11].

In recent years, there has been an increase in studies from hospital-based etiological surveillance, reflecting a growing awareness of the importance of HAdV as respiratory pathogens. A global study concluded that adenovirus infections accounted for 5–10% of respiratory infections in children and 1–7% in adults, and caused pneumonia in up to 20% of newborns and infants. In patients with severe HAdV pneumonia, the mortality rate may exceed 50% [12]. The positive detection rate of adenovirus was 3.9% and the mortality rate was 3% from 2004 to 2018 among inpatients hospitalized due to severe acute respiratory infection [13]. Studies in mainland China have shown that positive detection rate of adenovirus among ARTI patients was approximately 5.8%–13%, and the main affected groups were children and young adults [14]. Studies showed that the positive detection rate of adenovirus was 5.64% among hospitalized children with ARTI in Beijing from 2017 to 2018 and 6.9% in Zhejiang from 2018 to 2019 [15,16]. In general, there is still a lack of data on the HAdV prevalence and the predominant virus types responsible for ARTI sporadic outbreaks or epidemics in China.

Here we conduct a systematic review and meta-analysis of all published research articles on outbreak investigation and etiological surveillance of HAdV associated with cases of respiratory infection in China at the nation-wide level from 2009 to 2020 to evaluate the HAdV prevalence, virus types, seasonality, as well as to characterize patients' demographic and clinical data. This information might help to comprehensively understand the epidemic patterns of HAdV in China and support the adoption of targeted prevention and control measures.

## Materials and methods

This review was conducted according to the Preferred Reporting Items for Systematic Reviews and Meta-Analyses (PRISMA) statement (S1 PRISMA Checklist), and has been registered with the international prospective register of systematic reviews (PROSPERO) (International Prospective Register of Ongoing Systematic Reviews) (CRD42022303015) [17].

### Search strategy and selection criteria

Literature search was performed from the major databases including the PubMed database (https://pubmed.ncbi.nlm.nih.gov/), China National Knowledge Infrastructure (CNKI) (http://www.cnki.net/), Chongqing VIP Chinese Science and Technology Journal Database (CQVIP) (http://www.cqvip.com) and Wanfang databases (http://www.wanfangdata.com.cn/), with the keywords ('HAdV' OR 'adenovirus' [Title/Abstract]) AND ('respiratory' [Title/Abstract] OR 'pneumonia' [Title/Abstract]), AND ('China' OR 'the mainland of China' OR 'Chinese mainland' OR 'Taiwan' OR 'Hong Kong' OR 'Macau' OR 'Macao' [Title/Abstract]) (Table A in S1 Text). All the articles published between January 2009 and March 2021 were searched without language limitations.

We included studies of human infection with HAdV, across all settings (i.e., hospital, community, long-term care) and among all age groups (pediatric and adult patients). We included etiological surveillance studies and outbreak investigation, but excluded reviews, editorials, letters, case studies, randomized controlled trials and experimental studies. Studies were eligible if they explicitly described the total number of individuals tested and those that were positive for HAdV infections in humans. The following articles were excluded: (1) drug, vaccine trials, mechanism studies, animal experiments or reviews for HAdV; (2) etiological surveillance studies with sampling size <100 for laboratory test or HAdV positive detection <10; (3) describing cases imported from abroad after international travel; (4) lacking information about methods of laboratory diagnosis, specimens tested; (5) evaluations on laboratory methods for HAdV; (6) study period beyond the duration from 2009 to 2020 (Table B and C in S1 Text).

Titles and abstracts of the retrieved studies were screened using Endnote X9 independently by two reviewers (MCL and TTL) to identify studies potentially eligible for inclusion, and then the full texts were retrieved and independently assessed for eligibility. Discrepancies between reviewers were resolved by consensus or a third reviewer (QX). Studies potentially describing overlapping data were noted and the duplication were removed (e.g., same hospital and population during an overlapping time period).

## Data extraction and variable definition

One of the authors (MCL) extracted data from included studies using a standardized data collection form. The following variables were collected: reference ID, author, publication year, study sites, start and end dates, name(s) of healthcare facility; study design (etiological surveillance study, outbreak investigation), outbreak setting (school/daycare, healthcare comprised of hospitals and long-term care facilities, military camps, swimming pools), age group, patient population, mean or median age, gender proportion, laboratory test methods (molecular, serological) and type of HAdV, sample size, absolute number or rate of positive detection, presence of clinical symptoms or syndromes of patients if reported (Table D and E in S1 Text). For quality assurance, another two authors (QX, TW) randomly sampled 25% of recorded data to confirm accuracy and completeness.

For definition of outbreak event, all those recognized and reported outbreaks related to HAdV by health agencies were included. Otherwise, an outbreak event was defined as a number of clustered HAdV cases with a higher incidence than the average or expected incidence for a region where the cases occur [18]. All the events had to be laboratory confirmed, e.g., etiological pathogen determined to be HAdV by molecular methods (PCR) or serological methods (ELISA, IFA), while those outbreaks reporting suspected HAdV without laboratory confirmation for HAdV were not included in the analysis. For outbreak investigation, we extracted additional information regarding the exact date of outbreak, attack rate, numbers of primary cases and persons at risk, and number of secondary cases if available. For articles reporting more than one outbreak, data were separately extracted for each outbreak. For outbreaks reported in multiple publications, we included the one that reported more detailed data.

Four age groups were defined for comparison, including children (<5 years old), adolescents (5–17 years old), adults (18–59 years old), and the elderly (≥60 years old). When a study did not mention any age information, the all-age group was specified. Seven regions were defined according to the ecoclimatic characteristics, i.e., Northeast China, North China, Inner Mongolia-Xinjiang, Qinghai-Tibet, Southwest China, Central China, and South China [19].

## Meta-analysis

We performed the meta-analysis to evaluate demographic characteristics of patients, attack rate, or positive detection rate for HAdV. Briefly, the pooled proportion and 95% CI were estimated using the inverse variance combined with fixed effects or random effects models depending on the degree of the heterogeneity between studies. Heterogeneity was quantified using the statistic Higgin's $I^2$, when its value was greater than 50%, random effects model was used, otherwise, fixed effects model was applied [20]. We performed the meta-analysis to estimate the clinical manifestations that were related to different HAdV types, based on the 105 articles with a study size more than 20 patients and reporting the information of clinical manifestation. For those clinical manifestations which were reported only in one study, the proportion was calculated without a 95% CI estimated (S2 Text). All maps were produced by using the ArcGIS 10.7 software. The Meta program package in R 4.1.2 software was used to merge the rates and draw forest plots. All analyses were conducted with R 4.1.2 software.

## Results

### Temporal and spatial features of publications and patients

A total of 5,056 studies published from January 2009 to March 2021 were identified, 3,874 studies underwent title and abstract screening after duplicate removal, among which 1,329 were assessed via full-text screening. We included 950 studies (881 in Chinese and 69 in English) in the final analysis, comprised of 859 etiological surveillance studies involving 119,838 patients and 91 outbreak investigations involving 15,940 patients (Fig 1 and S1 List of References).

Of the 91 articles reporting 97 outbreak events, 68 (74.73%) were published between 2014–2019, with the highest number of articles published in 2014 (14 articles), followed by 2017 (13) (Fig 2A). Of the 859 etiological surveillance studies, 594 (69.15%) were published during 2015–2020, with the largest number published in 2015 (135 articles), followed by 2016 (103) (Fig 2C).

A comparable number of outbreaks took place in Northern and Southern China (Table 1). The geographic discrepancy of seasonal timing was shown for the outbreaks, with most of the outbreak events occurring in the winter season in Northern China (28/46), while a dual seasonal timing was observed in Southern China, at the turn of spring and summer and winter separately (44/51) (Fig 3A). The overall attack rate was estimated to be 15.91% (95% CI: 13.85–

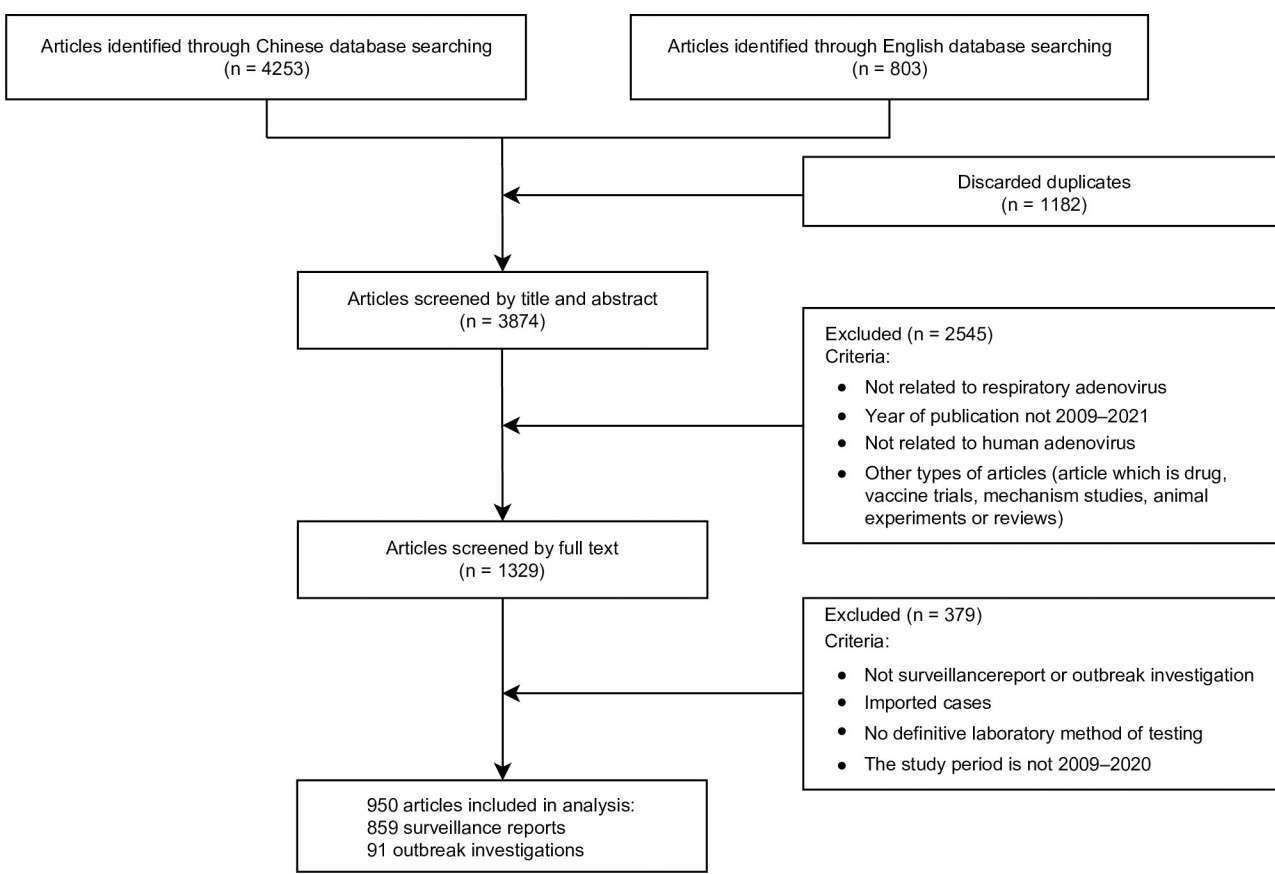

**Fig 1. Flow diagram of the literature review.** Literature search was performed from the major databases including the PubMed database (https://pubmed.ncbi.nlm.nih.gov/), China National Knowledge Infrastructure (CNKI) (http://www.cnki.net/), Chongqing VIP Chinese Science and Technology Journal Database (CQVIP) (http://www.cqvip.com) and Wanfang databases (http://www.wanfangdata.com.cn/).

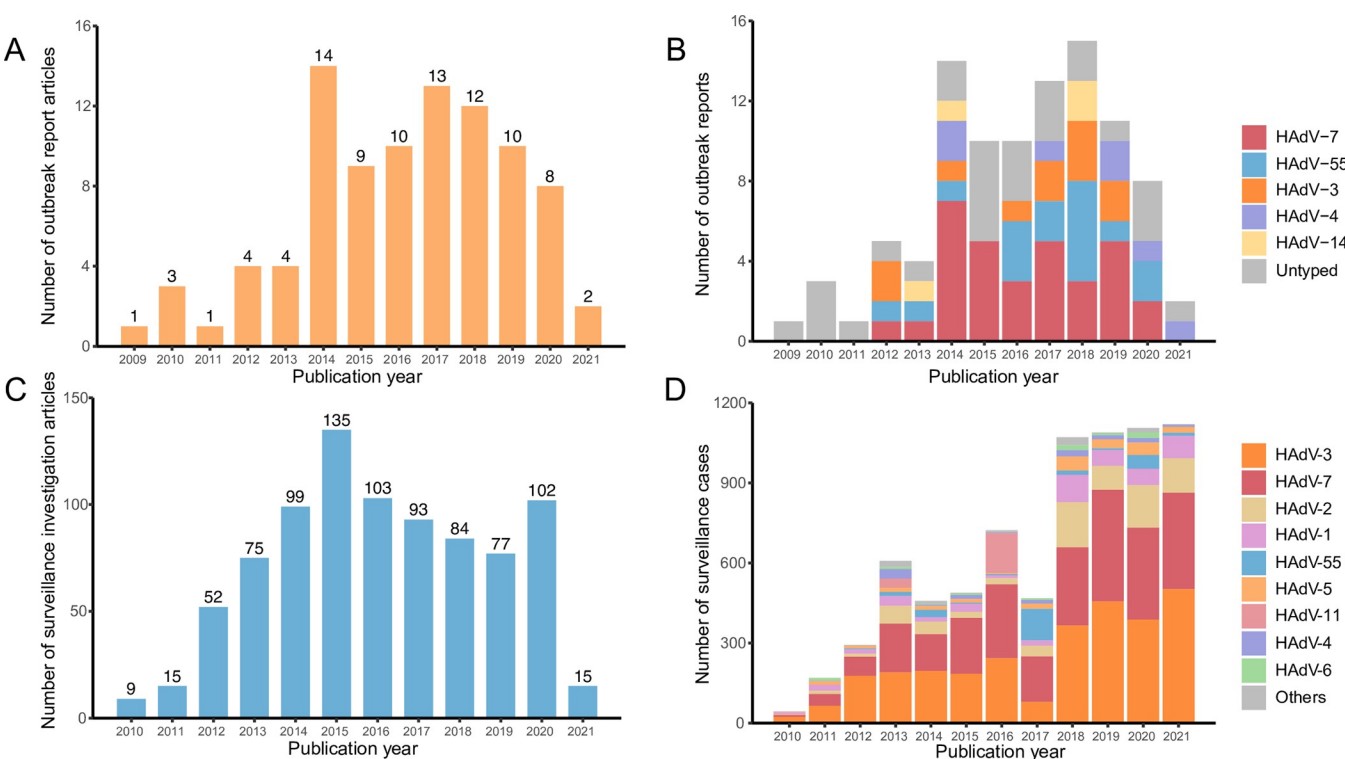

**Fig 2. Temporal pattern of reported outbreaks and etiological surveillance as well as the HAdV types in China.** (A) number of articles about reported outbreak events over publication year; (B) number of reported outbreaks according to each HAdV type group over publication year; (C) number of articles about etiological surveillance of HAdV over publication year; (D) number of patients reported in etiological surveillance according to each HAdV type group over publication year. Other in the panel D indicate those rarely seen HAdV types, including HAdV-21 (30 patients), HAdV-57 (18), HAdV-31 (5), HAdV-50 (2), HAdV-12 (1), HAdV-35 (1), and HAdV-104 (1). Deadline for literature search is March 2021.

17.98). A higher attack rate was observed in Northern China than in Southern China (19.01%, 95% CI: 14.42–23.60 versus 13.53%, 95% CI: 11.28–15.77). When the outbreak settings were compared, the highest attack rate was observed in military camps (23.55%, 95% CI:18.02–29.07), followed by swimming pools (22.47%, 95% CI: 12.49–32.45), hospitals (19.75%, 95% CI: 8.64–30.86) and schools (6.19%, 95% CI: 4.92–7.46) (Fig 3B). In contrast with the outbreak events, the etiological surveillance studies among ARTI patients reported comparable interregional positive rate (4.13%, 95% CI: 3.95–4.31 in Southern China, 4.15%, 95% CI: 3.95–4.34 in Northern China), and the overall positive detection rate was estimated to be 4.21% (95% CI, 4.07–4.34) (Fig 3C and Table F–J in S1 Text).

## Demographic characteristics of patients

The highest number of outbreak events was observed in the adult group (47), followed by adolescents (43) and children (6). The highest attack rate was shown in adults (23.56%, 95% CI: 18.31–28.80), followed by children (12.78%, 95% CI: 7.95–17.62) and adolescents (6.52%, 95% CI: 5.50–7.55). An age pattern of HAdV infection was shown from the etiological surveillance data, with higher positive rate observed in children (4.04%, 95% CI: 3.76–4.31) and adolescents (4.45%, 95% CI: 2.55–6.36) than those of the two older groups (Table 1). Seasonal pattern differed between age groups. For children and adolescents, over half of the outbreaks occurred in autumn and spring (29 of 49 outbreaks), while for adult groups, most outbreaks occurred in winter (38/47) (Fig 3D and Table K in S1 Text).

**Table 1. Attack rate and positive detection rate of HAdV by areas, seasons, patients' ages, settings and virus types based on meta-analysis.**

| | Outbreak events | | | Etiological surveillance | | |
|---|---|---|---|---|---|---|
| | Number of articles (No. of outbreaks) | Cases | Attack rate by meta-analysis % (95% CI)* | Number of articles | Cases | Positive detection rate by meta-analysis % (95% CI) |
| **Number** | 91 (97) | 15,940 | 15.91 (13.85, 17.98) | 859 | 119,838 | 4.21(4.07, 4.34) |
| **Mortality** | 4 (4) | 4 | - | 9 | 37 | - |
| **Areas**** | | | | | | |
| **Northern** | 44 (46) | 9,182 | 19.01 (14.42, 23.60) | 283 | 19,955 | 4.15 (3.95, 4.34) |
| **Southern** | 48 (51) | 6,758 | 13.53 (11.28, 15.77) | 584 | 97,162 | 4.13 (3.95, 4.31) |
| **Season** | | | | | | |
| **Spring** | 19 (20) | 899 | 5.92 (4.47, 7.36) | 39 | 2,047 | 5.21 (4.35, 6.07) |
| **Summer** | 16 (17) | 440 | 16.76 (12.47, 21.04) | 39 | 1,780 | 4.64 (3.82, 5.47) |
| **Autumn** | 16 (17) | 394 | 5.71 (3.41, 8.01) | 36 | 1,278 | 3.62 (2.95, 4.29) |
| **Winter** | 40 (43) | 8,453 | 22.65 (16.33, 28.96) | 37 | 2,042 | 3.54 (2.83, 4.24) |
| **Age** | | | | | | |
| **Children** | 6 (6) | 121 | 12.78 (7.95, 17.62) | 174 | 10,671 | 4.04 (3.76, 4.31) |
| **Adolescents** | 40 (43) | 2,042 | 6.52 (5.5, 7.55) | 8 | 531 | 4.45 (2.55, 6.36) |
| **Adults** | 44 (47) | 13,734 | 23.56 (18.31, 28.8) | 28 | 1,219 | 3.44 (2.76, 4.12) |
| **The elderly** | 0 (0) | 0 | | 9 | 229 | 2.81 (1.84, 3.78) |
| **All-age groups** | 1 (1) | 43 | 25.29 | 644 | 107,188 | 4.29 (4.12, 4.45) |
| **Settings** | | | | | | |
| **School** | 41 (43) | 3,386 | 6.19 (4.92, 7.46) | 0 | 0 | |
| **Military camp** | 37 (40) | 11,849 | 23.55 (18.02, 29.07) | 0 | 0 | |
| **Hospital** | 5 (5) | 142 | 19.75 (8.64, 30.86) | 859 | 119,838 | 4.21(4.07, 4.34) |
| **Swimming pool** | 9 (9) | 563 | 22.47 (12.49, 32.45) | 0 | 0 | |
| **Types†** | | | | | | |
| **HAdV-1** | 0 (0) | 0 | - | 40 | 466 | 6.70 (5.39, 8.01) |
| **HAdV-2** | 0 (0) | 0 | - | 44 | 772 | 8.90 (7.31, 10.50) |
| **HAdV-3** | 9 (11) | 711 | 5.55 (2.63, 8.48) | 60 | 2,869 | 32.73 (22.13, 43.34) |
| **HAdV-4** | 7 (7) | 150 | 8.75 (0.00, 21.30) | 30 | 137 | 2.07 (1.41, 2.74) |
| **HAdV-5** | 0 (0) | 0 | - | 38 | 249 | 3.55 (2.78, 4.32) |
| **HAdV-6** | 0 (0) | 0 | - | 21 | 83 | 1.97 (1.23, 2.70) |
| **HAdV-7** | 30 (32) | 7,048 | 22.32 (14.78, 29.86) | 59 | 2,518 | 27.48 (17.04, 37.91) |
| **HAdV-11** | 0 (0) | 0 | - | 4 | 186 | 16.14 (1.89, 30.40) |
| **HAdV-14** | 3 (4) | 92 | 8.83 (6.31, 11.35) | 12 | 43 | 2.01 (0.92, 3.09) |
| **HAdV-21** | 0 (0) | 0 | - | 7 | 30 | 0.87 (0.17, 1.57) |
| **HAdV-31** | 0 (0) | 0 | - | 4 | 5 | 0.33 (0.00, 0.73) |
| **HAdV-55** | 16 (16) | 4,043 | 27.18 (19.16, 35.20) | 25 | 258 | 4.70 (3.40, 6.00) |
| **HAdV-57** | 0 (0) | 0 | - | 8 | 18 | 1.01 (0.51, 1.50) |

* 53 outbreaks recording attack rate were included in the total; 2, 26, 24 and 1 outbreaks recording attack rate were included in the children, adolescents, adults, and all age groups, respectively; 13, 8, 6 and 25 outbreaks recording attack rate were included in the spring, summer, autumn and winter groups, respectively; 25 and 28 outbreaks recording attack rate were included in the Northern and Southern groups, respectively; 22, 2, 6 and 23 outbreaks recording attack rate were included in the military, hospital swimming pool and school groups, respectively; 4, 2, 17, 1 and 9 outbreaks recording attack rate were included in the HAdV-3, HAdV-4, HAdV-7, HAdV-14 and HAdV-55 groups, respectively.

** Of the 848 articles mentioning locations, 19 mentioned both Northern and Southern China.

†HAdV type with case number > 5 were included, other rarely seen types including HAdV-50 (2 cases), HAdV-12 (1), HAdV-35 (1), and HAdV-104 (1).

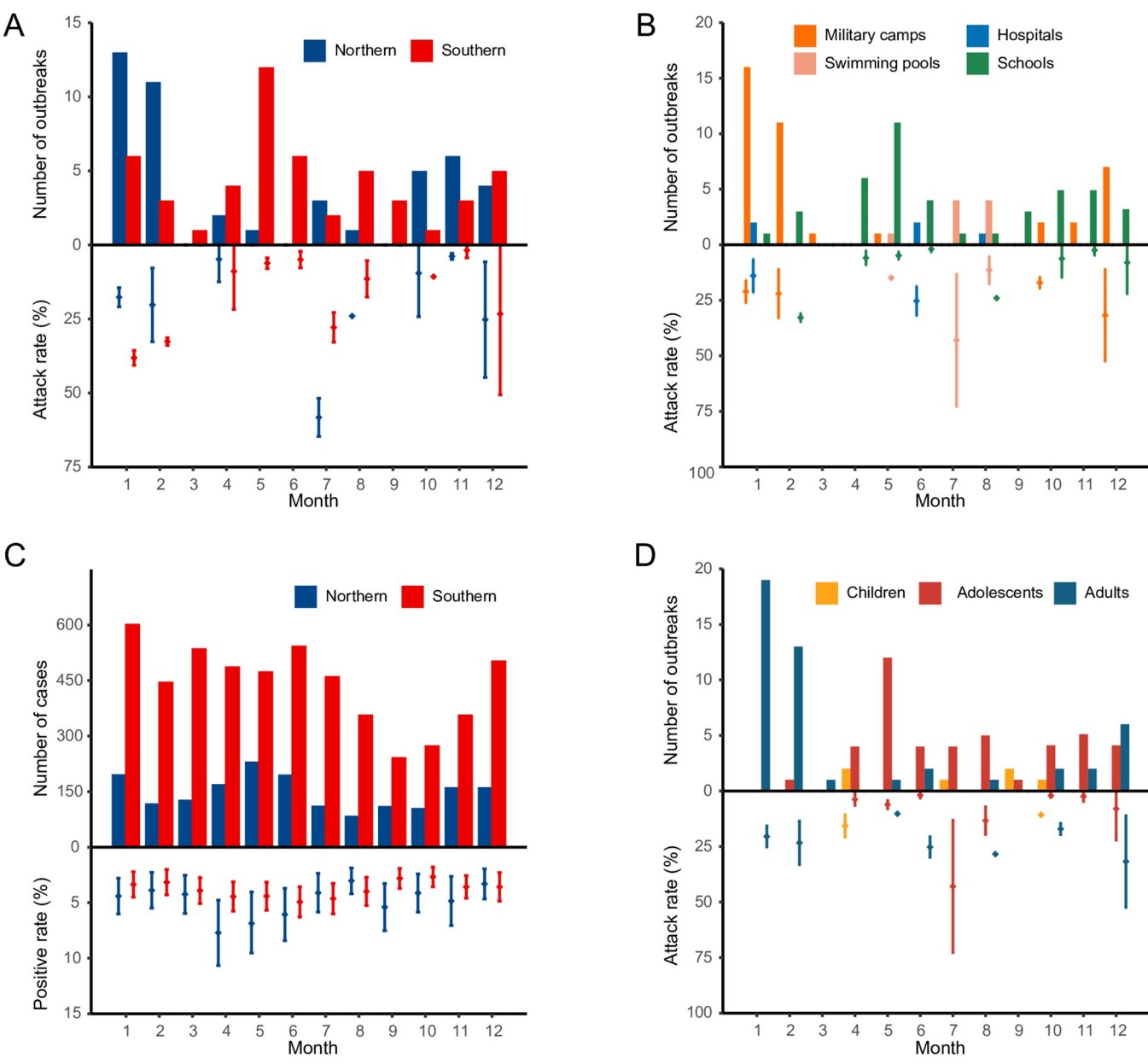

**Fig 3. Attack rate of HAdV in outbreaks and positive detection rate of HAdV in etiological surveillance over months based on the meta-analysis.** (A) attack rate by region; (B) attack rate by settings; (C) positive detection rate by regions; (D) attack rate by age groups. The bars in panels A, B, and D indicate the number of outbreaks, and the intervals indicate the attack rate and 95% CI. The bars on panel C indicate the case number in etiological surveillance, and the intervals indicate the positive detection rate and 95% CI.

Among 97 outbreaks that reported settings, the highest number was reported in schools (44.33%, 43/97), followed by military camps (41.24%, 40/97), swimming pools (9.28%, 9/97), and hospitals (5.15%, 5/97). The seasonality depended largely on the outbreak settings, with most of the military camps outbreaks occurring in winter (85.00%, 34/40), and most swimming pool outbreaks occurring in summer (88.89%, 8/9), while school outbreaks usually occurred in spring and autumn during school terms as expected (69.77%, 30/43). A total of 41 deaths with confirmed HAdV infection were reported from 13 articles, with an overall case fatality rate of 0.03% (Table 1).

## Temporal and geographic pattern of HAdV types in China

Sequence information was available for 70 outbreak events involving 12,044 cases and 67 etiological surveillance studies involving 7,639 cases. The most common type responsible for outbreak events was HAdV-7, accounting for 45.71% (32/70) of the total number of outbreaks for which HAdV typing was performed, followed by HAdV-55 (16/70), HAdV-3 (11/70), HAdV-4 (7/70), and HAdV-14 (4/70). HAdV-7 had the highest number of reported outbreaks in 2014, 2015 and 2017, while HAdV-55 (5) had the highest proportion of reported outbreaks in 2018 (Fig 2B). The case numbers involved in outbreaks caused by HAdV-7, HAdV-55 and HAdV-3 were 7,048, 4,043 and 711, respectively, based on which the attack rate of HAdV-55 was estimated to be 27.18% (95% CI: 19.16–35.20), which was significantly higher than that of HAdV-7 (22.32%, 95% CI: 14.78–29.86) and HAdV-3 (5.55%, 95% CI: 2.63–8.48) (Table 1). A difference of the predominant HAdV type in outbreak events was shown between Northern China and Southern China. Among all the outbreaks reporting HAdV types and study sites, HAdV-55 (12 events involving 3,841 cases) and HAdV-7 (17 events involving 2,614 cases) were predominant in Northern China, while HAdV-7 (15 events involving 4,434 cases), HAdV-3 (seven events involving 640 cases), and HAdV-55 (four events involving 202 cases) were predominant in Southern China (Fig 4A). According to the ecological regions of China, we found that HAdV-55 was the dominant HAdV type for outbreak events in four regions including Inner Mongolia-Xinjiang, North China, Qinghai-Tibet, and South China, while HAdV-7 was the dominant type in the others including Northeast China and Central China, except for one region(Southwest China) without reporting outbreaks with HAdV typing information (Table L in S1 Text).

The predominant HAdV types differed from those in outbreak events. The most common type was HAdV-3 (2,869 cases), followed by HAdV-7 (2,518), HAdV-2 (772), HAdV-1 (466), HAdV-55 (258), and HAdV-5 (249). Both HAdV-3 and HAdV-7 had been shown an increasing pattern in the number of reported cases during the study period (Fig 2D). The positive rate of HAdV-3 was determined as 32.73% (95% CI: 22.13–43.34), which was significantly higher than that of HAdV-7 (27.48%), HAdV-11 (16.14%), HAdV-2 (8.90%), HAdV-1 (6.70%), HAdV-55 (4.70%), and. HAdV-5 (3.55%). Both HAdV-3 and HAdV-7 were the common types identified in Northern and Southern China (Fig 4B and Table 1).

Among 70 outbreaks providing information on age and HAdV type, HAdV-7 (25) and HAdV-55 (15) were responsible for all the 40 outbreaks reported in the adult group; HAdV-7 (3) and HAdV-4 (3) were determined in the six outbreaks reported from children, while a higher diversity of types was observed in the 24 outbreaks reported in the adolescent group, including HAdV-3 (11), HAdV-4 (4), HAdV-7 (4), HAdV-14 (4), and HAdV-55 (1) (Fig 4C). Among the 70 outbreaks with reported settings and types, HAdV-7 (21) and HAdV-55 (12) were responsible for all the 33 outbreaks in military camps, which involved 5,111 and 3,674 cases, respectively. HAdV-55 (3 outbreaks) and HAdV-7 (2) were responsible for the five outbreaks in hospitals, with 113 and 29 cases involved respectively. HAdV-3 (2 outbreaks), HAdV-4 (1), and HAdV-7 (2) were determined to be responsible for the five swimming pool outbreaks, with 199, 147, and 9 cases involved respectively. Five HAdV types were responsible for the 28 outbreaks in schools, including HAdV-3 (9), HAdV-7 (7), HAdV-4 (6), HAdV-14 (4), and HAdV-55 (2), with 1, 761 cases infected with HAdV-7 and 512 cases with HAdV-3 infection (Fig 4D).

## Clinical manifestations

Altogether, 105 articles with HAdV cases greater than 20 were included in the analysis of clinical manifestations. Among children with HAdV infection, cough was the most prevalent

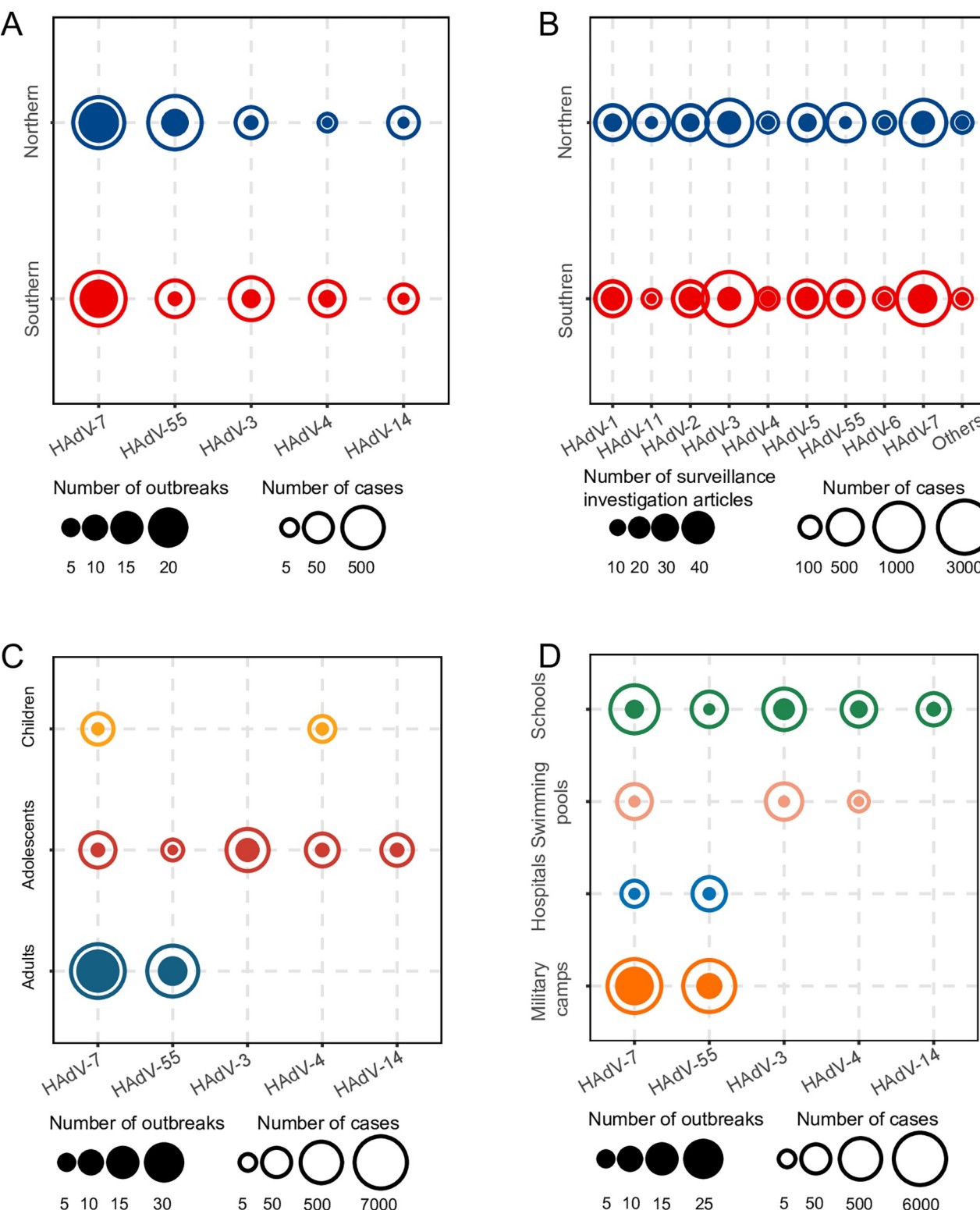

**Fig 4. Comparison of number of outbreak events or etiological surveillance and case number with HAdV types by region, age, and setting.** (A) outbreaks in different regions; (B) etiological surveillance in different regions; (C) outbreaks in different age groups; (D) outbreaks in different settings. Solid circles in panels A, C, and D indicate the number of outbreaks, and hollow circles indicate the number of cases. Solid circles in panel B indicate the number of etiological surveillances, and hollow circles indicate the number of cases.

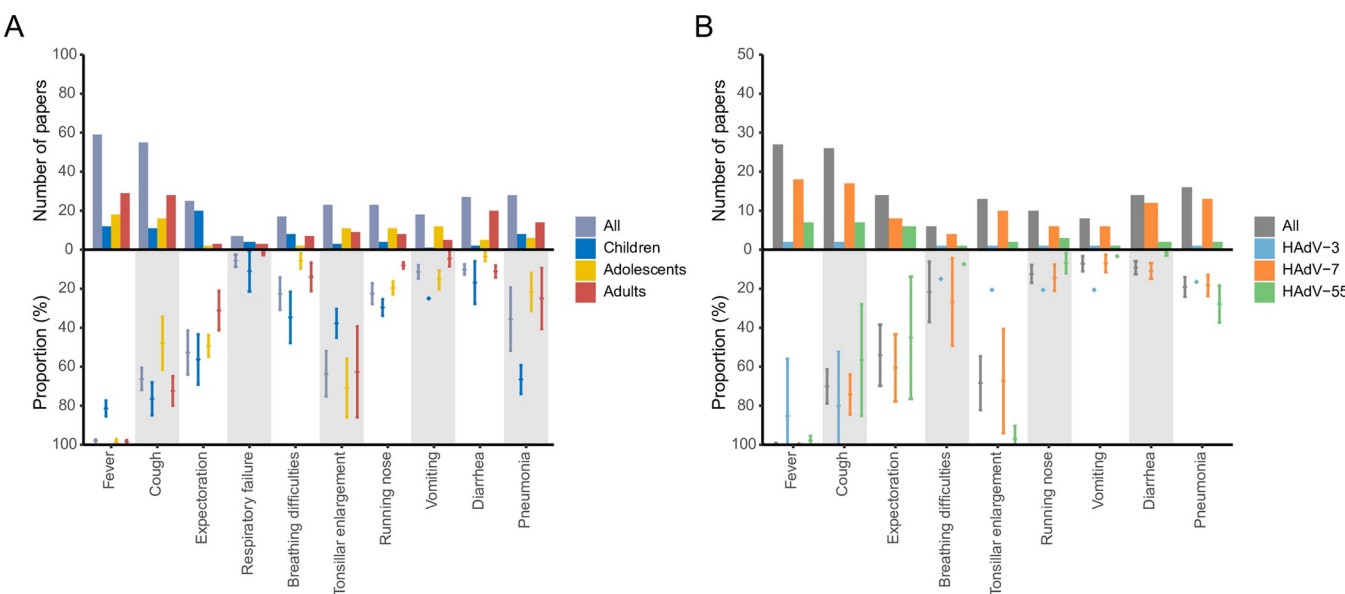

**Fig 5. Clinical manifestations of HAdV infection by age groups and virus types.** (A) for different age groups; (B) for different virus types. The bars indicate the number of articles, and the intervals indicate the proportion of patients with the clinical manifestation and 95% CI based on the meta-analysis.

symptom, identified in 76.47% of the cases, followed by pneumonia (66.56%), expectoration (56.28%), respiratory failure (11.05%), breathing difficulties (34.69%), running nose (29.58%), vomiting (25.00%), and diarrhea (16.87%). Among adolescents, tonsillar enlargement (70.89%), pneumonia (21.69%) and vomiting (15.43%) were frequently seen. Among adults, less diverse clinical presentations were seen, including expectoration (31.19%), running nose (8.05%) and vomiting (4.60%) (Fig 5A, Table M in S1 Text).

Compared with the infection of HAdV-55 and HAdV-7, HAdV-3 infection was related to higher frequency of cough (79.91%), running nose (20.59%) and vomiting (20.59%). HAdV-55 infection was related to higher frequency of tonsillar enlargement (96.91%), pneumonia (27.91%), while lower occurrence of cough (56.61%), expectoration (45.10%), running nose (6.90%), and vomiting (3.37%), diarrhea (1.50%), compared with the infection of HAdV-3 and HAdV-7. HAdV-7 infection was related to higher presence of expectoration (60.59%), breathing difficulties (26.80%), compared with the infections of HAdV-3 and HAdV-55 (Fig 5B and Fig A in S1 Text).

## Discussion

The current study provided a most comprehensive and up to date estimation on the attack rate/positive rate of HAdV as well as the dominant virus types, that differed across age, setting and seasons. Attack rates were significantly higher in outbreaks related to military camps and in the winter season. HAdV-55, HAdV-7 and HAdV-3 were the major causative agents in the outbreak events. Outbreaks in military camps were more likely to be associated with HAdV-7 and HAdV-55, while outbreaks in other setting were associated with more HAdV genotypes, particularly for the school outbreaks. This was also in line with the higher diversity of virus types in the adolescent group, who constituted the major part of school outbreaks. These virus typing results contrasted with the etiological surveillance studies where HAdV-3 and HadV-7 were the most frequently determined when an all-age group was studied. The current surveillance finding was highly consistent with the prior epidemiological investigation performed in

China, where HAdV-3 and HAdV-B7 were the most frequently detected among acute respiratory distress syndrome patients [21–24].

We found that school-related outbreaks were associated with the lowest attack rate, suggesting either a less confined space that limited the transmission or the lower transmission capacity relate to different HAdV types. We also revealed a different situation from that shown in other countries, for example, in the American troops, HAdV-4, HAdV-B7 and HAdV-B14 act as the predominant strains sequentially isolated from the outbreaks [7,25–27]. HAdV vaccination program against the two-outbreak-related virus types in this high-risk population is urgently needed. Our meta-analysis showed that overall attack rate of adenovirus infection in outbreaks was estimated at 15.91% (95% CI: 13.85–17.98), which is comparable with the outbreak among college students in Pennsylvania, USA, of which a 15% (44/288) attack rate was reported based on the test of nasopharyngeal swabs for HAdV [28].

Our study for the first time revealed a clear seasonal pattern for the outbreak events. The school outbreaks occurred mostly at the beginning of new school years, while the military outbreaks mainly occurred at the recruit training seasons. A Korean study from 2013 to 2018 showed that HAdV was the most frequently detected respiratory virus in military recruits (6,646/14,630, 45.4%) [29]. HAdV has been implicated in over half of the febrile respiratory illness cases reported at recruit training center clinics.

A total of 17 virus types involving respiratory adenovirus infection have been reported in China, of which HAdV-3, HAdV-7 and HAdV-55 were predominant. Outbreaks were dominated by HAdV-7 and HAdV-55, while other types included HAdV-3, HAdV-4 and HAdV-14. The main virus types of outbreaks differ in different settings, with school outbreaks having more types, including HAdV-7, HAdV-55, HAdV-3, HAdV-4 and HAdV-14, with HAdV-3 and HAdV-7 predominating; swimming pool outbreaks having mainly HAdV-3, HAdV-7 and HAdV-4, mainly in summer; hospital outbreaks and military outbreaks having mainly HAdV-7 and HAdV-55. The virus types of outbreaks vary by age groups, HAdV-7 and HAdV-4 are dominant in children, HAdV-7 and HAdV-55 are dominant in adults, and more virus types are shown in adolescents, including HAdV-7, HAdV-55, HAdV-3, HAdV-4, and HAdV-14. There are more virus types detected in etiological surveillance, including HAdV-3, HAdV-7, HAdV-2, HAdV-1, HAdV-55, HAdV-5, etc. The most common types are HAdV-3 and HAdV-7 with an increasing trend along the years, which may be related to both the increasing number of infections and rising levels of surveillance. Differences in the main HAdV types in outbreak investigations and etiological surveillance reports suggest that HAdV-55 is more infectious and more likely to lead to outbreaks, whereas HAdV-3 is less infectious than HAdV-55 and HAdV-7 and is easily detected during hospital-based surveillance. This may be because HAdV-55 is more symptomatic, more contagious and more likely to be reported. A study has shown that HAdV-7 replicates more robustly than HAdV-3, and promotes an exacerbated cytokine response, causing a more severe airway inflammation [30]. In our study, the mortality rate of adenovirus infection in patients with respiratory infections was about 0.03%, while a study in Rio Grande Do Sul, Brazil showed a higher mortality rate (3%) among hospitalized patients with severe acute respiratory infection, which might be caused by a lower proportion of patients with severe respiratory infection in our study [13].

Overall, adenovirus circulates throughout the year, with slightly higher numbers reported during the summer and winter. Previous studies have shown that HAdV detection rates are positively associated with the monthly mean temperature and sunshine duration, and negatively associated with wind speed [31]. In our study, the main prevalent adenovirus types were HAdV-3, HAdV-7 and HAdV-55 in China, with other more frequent types such as HAdV-2, HAdV-4, HAdV-1 and HAdV-5. The main outbreak sites were schools and the military, with higher attack rates in the north than in the south. The prevalent virus typing varies between

countries and regions. Species B, C and D are the most common adenoviruses worldwide, of which species B and C could cause respiratory infections. For species B, a low prevalence of HAdV-7, HAdV-11 and HAdV-35 was reported globally, however, it is high prevalence for HAdV-35 in some counties from Africa with a positive detection rate of about 20% among HIV-infected patients, and a high prevalence of HAdV-7 was reported in China, United States and Belgium. For species C, HAdV-5 was the widely distributed species, and HAdV-2 usually had a high positive detection rate among healthy individuals and HIV-infected patients in China and developed countries [5]. An epidemiologic study based on HAdV molecular typing was conducted in the Korean military from January 2013 to April 2014, and HAdV-55 (42.0%) was the most frequently identified strain, followed by HAdV-4 (13.0%), HAdV-5 (1.4%), and HAdV-6 (1.4%) [32]. HAdV-55 is a recently identified pathogen, which evolved from recombination between adenovirus 11 and 14 [33,34]. It was initially described as serotype 11a and was later re-labeled as HAdV-55 because of its recombinant genome [33], which is mainly found in China and Korea [35].

The clinical presentation varies by age group and by virus type. Most adenovirus respiratory infections are light to moderate and self-limited; however, sometimes they may cause life-threatening conditions, comorbidities, and serious sequelae. The rate of pneumonia is higher in children than in other age groups. In addition, the children are more likely to have breathing difficulties. Attention should be paid to the occurrence of adenovirus pneumonia in children. The main symptoms are fever and tonsil enlargement in adolescents while the main symptoms are cough and tonsil enlargement in adults. Adolescents and adults have a stronger immune system and the symptoms are mainly mild. Compared to other virus types, HAdV-55 is more likely to cause pneumonia and has a high prevalence in the military camps, so attention should be paid to the prevention and control of adenovirus in the military camps.

There were two main limitations to this study. First, inherent to systematic reviews, our study was influenced by publication bias. Most outbreaks are reported by passive surveillance which may not be as comprehensive as in active surveillance, and are also subject to reporting biases. Secondly, as our data came from different studies, and did not have the same variables. This may have increased the likelihood of misclassification bias, also limited the number of variables that can be used for analysis.

Despite of these limitations, we have disclosed the prevalence of human infection with respiratory adenovirus and the major genotypes that differed over time, by location, and by demographical characteristics, e.g., patients' age. Comparing and contrasting the features across diverse settings can help to attain an enhanced epidemiological and clinical understanding of human infections with different HAdV types, and thus enhancing the accuracy of HAdV surveillance systems.

## Supporting information

**S1 PRISMA Checklist. A complete checklist of risk of bias according to the Preferred Reporting Items for Systematic Reviews and Meta-Analyses (PRISMA) statement.**
(DOCX)

**S1 Text. All data used to draw the conclusions are outlined in the manuscript, including the literature screening process, the meta-analysis process, and the regional distribution of the different adenovirus types.**
(DOCX)

**S2 Text. The forest plots used in the meta-analysis.**
(DOCX)

**S1 List of References. 950 references used in the systematic review and meta-analysis.**
(XLSX)

## Acknowledgments

The authors would like to thank all patients who contributed data and clinicians who collected the data.

## Author Contributions

**Conceptualization:** Wei Liu, Li-Qun Fang.

**Data curation:** Mei-Chen Liu, Qiang Xu, Ting-Ting Li, Tao Wang, Bao-Gui Jiang, Chen-Long Lv.

**Formal analysis:** Mei-Chen Liu, Ting-Ting Li.

**Funding acquisition:** Wei Liu.

**Investigation:** Mei-Chen Liu, Qiang Xu, Ting-Ting Li, Tao Wang, Bao-Gui Jiang, Chen-Long Lv.

**Methodology:** Qiang Xu, Tao Wang, Bao-Gui Jiang, Chen-Long Lv.

**Project administration:** Wei Liu, Li-Qun Fang.

**Resources:** Wei Liu, Li-Qun Fang.

**Software:** Wei Liu, Li-Qun Fang.

**Supervision:** Wei Liu, Li-Qun Fang.

**Validation:** Qiang Xu, Tao Wang, Bao-Gui Jiang, Chen-Long Lv, Xiao-Ai Zhang.

**Visualization:** Qiang Xu, Tao Wang, Bao-Gui Jiang, Chen-Long Lv, Xiao-Ai Zhang.

**Writing – original draft:** Mei-Chen Liu, Qiang Xu.

**Writing – review & editing:** Wei Liu, Li-Qun Fang.

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
