## [Decision Letter · Decision Letter 0]

18 Dec 2022

Dear Dr Fang,

Thank you very much for submitting your manuscript "Human infection with respiratory adenovirus in China: a systematic review and meta-analysis" for consideration at PLOS Neglected Tropical Diseases. As with all papers reviewed by the journal, your manuscript was reviewed by members of the editorial board and by several independent reviewers. In light of the reviews (below this email), we would like to invite the resubmission of a significantly-revised version that takes into account the reviewers' comments. 

We cannot make any decision about publication until we have seen the revised manuscript and your response to the reviewers' comments. Your revised manuscript is also likely to be sent to reviewers for further evaluation.

Sincerely,

Ali Rostami

Academic Editor

Elvina Viennet

Section Editor

Reviewer's Responses to Questions

**Key Review Criteria Required for Acceptance?**

**Methods**

-Are the objectives of the study clearly articulated with a clear testable hypothesis stated?

-Is the study design appropriate to address the stated objectives?

-Is the population clearly described and appropriate for the hypothesis being tested?

-Is the sample size sufficient to ensure adequate power to address the hypothesis being tested?

-Were correct statistical analysis used to support conclusions?

-Are there concerns about ethical or regulatory requirements being met?

Reviewer #1: Title

It would be good to clarify if this article is a prevalence study or not.

Abstract

Methods should be written in more detail about the study design, sampling, and clinical manifestation.

Introduction

1. The introduction needs to be improved by adding some information about the global prevalence, mortality, morbidity, and burden of HAdV both worldwide and in China.

2. Page 4, line 75: “Here we conduct a systematic review of all published research...” isn't that a systematic review and meta-analysis? 

Methods

1. Page 5, line 109: Part 6 of the exclusion criteria says “Study period beyond the duration 2009-2020” while the authors indicated data extraction was conducted between Jan 2009 and Mar 2021. Also, in Appendix1 study period is 2009-2021 but in table S3 it is from 2009 to 2020. Please explain the difference.

2. In appendix1 figure S1, I can't see the distribution of adenovirus typing pie chart for southwest China on the map! 

3. There is not enough information about how the quality assessment was done! Please state the exact method that has been used, provide a complete checklist of risk of bias as an appendix, and address the references.

4. Page 6, lines 131-133: “an outbreak event was defined...” is there any reference for this definition?

5. Please report sensitivity and specificity of laboratory determination methods that have been used.

Reviewer #2: The hypothesis and objectives of the work are clearly stated.

The selection of the articles is carried out properly.

Statistical analyzes are adequate.

Reviewer #3: The study is a systematic review and meta-analysis about human adenovirus (HAdV) infection associated with respiratory disease in China. The methodology is in accordance with a systematic review and met analysis article.

**Results**

-Does the analysis presented match the analysis plan?

-Are the results clearly and completely presented?

-Are the figures (Tables, Images) of sufficient quality for clarity?

Reviewer #1: Results

There could have been data from HAdV vaccination in China among all age groups and a proper comparison of HAdV incidence between vaccinated and non-vaccinated individuals.

There could have been information about the most common type of HAdV separated by year from fig2B and fig2C.

Reviewer #2: The results section describes very well the analysis carried out considering three different age ranges, and important to describe.

The graphs are well designed and display the results in an easy to view manner.

The comparative table is correct and presents valuable information comparing outbreaks and surveillance.

Reviewer #3: A total of 5056 studies were identified, of which 950 articles met the inclusion criteria and were analyzed in the study. The study analyzed both articles from outbreak studies as well as from surveillance studies.

Presentations of results could be improved, especially concerning use of expressions and words not well explained in the text, grammar errors that make the text hard to understand, including in Figure legends.

**Conclusions**

-Are the conclusions supported by the data presented?

-Are the limitations of analysis clearly described?

-Do the authors discuss how these data can be helpful to advance our understanding of the topic under study?

-Is public health relevance addressed?

Reviewer #1: Discussion

Page 17 line 3: “The prevalent typing varies...” presented common HAdV types in the United States from 2003-2016. It would be better if authors used the same time period for comparison as the current study. And please mention the most common types globally not just one country like the USA.

Reference

Please add journal issue after volume number.

Reference order could have been reported by year, but it looks fine this way.

Minor issues

There is no line number after page 10.

Page 11, first paragraph line 5: rat � rate

Reviewer #2: The discussion is correct, with updated bibliography.

The results support the discussion and the conclusion reached by the authors.

The limitations of the study are clearly stated in the discussion.

Reviewer #3: The study discusses the analyses performed and its findings, comparing with studies from some other countries. I suggest including a discussion about mortality associated with HAdV infection, comparing findings of the study (0.03 mortality) with other studies, for example the study by Pscheidt et al. (2020 - https://doi.org/10.1002/rmv.2189) that found 3% fatalities among patients hospitalized with severe respiratory infection and who were HAdV-positive.

**Editorial and Data Presentation Modifications?**

Reviewer #1: (No Response)

Reviewer #2: I recommend the acceptance of the evaluated work.

Reviewer #3: (No Response)

**Summary and General Comments**

Reviewer #1: The manuscript “Human infection with respiratory adenovirus in China: a systematic review and meta-analysis” is a review of 950 articles that describes Human adenovirus (HAdV) prevalence, predominant types, clinical manifestation among different age groups and settings, national wide. The objective is interesting, although it needs some revisions before acceptance.

Reviewer #2: The manuscript entitled “Human infection with respiratory adenovirus in China: a systematic review and metaanalysis”, authored by Mei-Chen Liu et al, described the epidemiological and clinical features of HAdV infections in China, from January 2009 to March 2021. In addition, the genetic and epidemiological characteristics of HAdVs were investigated. 

I want to emphasize that the work is very well written and concisely covers all the epidemiological characteristics of HAdV respiratory infections. 

The work carried out is a very important contribution to the knowledge of the classical and molecular epidemiology of HAdV in China and also provides valuable information worldwide.

Reviewer #3: The study is interesting and is worth publication, however thorough English revision is necessary. I recommend revision by a native English speaker.

Main comments and corrections were made in the pdf file, attached.

PLOS authors have the option to publish the peer review history of their article (what does this mean?). If published, this will include your full peer review and any attached files.

Reviewer #1: No

Reviewer #2: No

Reviewer #3: No
---

## [Decision Letter · Decision Letter 1]

5 Feb 2023

Dear Dr Fang,

Thank you very much for submitting your manuscript "Prevalence of human infection with respiratory adenovirus in China: a systematic review and meta-analysis" for consideration at PLOS Neglected Tropical Diseases. As with all papers reviewed by the journal, your manuscript was reviewed by members of the editorial board and by several independent reviewers. The reviewers appreciated the attention to an important topic. Based on the reviews, we are likely to accept this manuscript for publication, providing that you modify the manuscript according to the review recommendations. 

Sincerely,

Ali Rostami

Academic Editor

Elvina Viennet

Section Editor

Reviewer's Responses to Questions

**Key Review Criteria Required for Acceptance?**

**Methods**

-Are the objectives of the study clearly articulated with a clear testable hypothesis stated?

-Is the study design appropriate to address the stated objectives?

-Is the population clearly described and appropriate for the hypothesis being tested?

-Is the sample size sufficient to ensure adequate power to address the hypothesis being tested?

-Were correct statistical analysis used to support conclusions?

-Are there concerns about ethical or regulatory requirements being met?

Reviewer #1: (No Response)

Reviewer #2: The objectives and design of the study are clear.

The statistical analysis performed was adequate.

Reviewer #3: Adequate.

**Results**

-Does the analysis presented match the analysis plan?

-Are the results clearly and completely presented?

-Are the figures (Tables, Images) of sufficient quality for clarity?

Reviewer #1: (No Response)

Reviewer #2: The results are clear and correctly presented, as are the tables and figures.

Reviewer #3: Adequate.

**Conclusions**

-Are the conclusions supported by the data presented?

-Are the limitations of analysis clearly described?

-Do the authors discuss how these data can be helpful to advance our understanding of the topic under study?

-Is public health relevance addressed?

Reviewer #1: (No Response)

Reviewer #2: The conclusions are supported by the results.

Study limitations are presented.

Reviewer #3: Adequate.

**Editorial and Data Presentation Modifications?**

Reviewer #1: (No Response)

Reviewer #2: Accept

Reviewer #3: Accept after minor corrections.

**Summary and General Comments**

Reviewer #1: (No Response)

Reviewer #2: The manuscript entitled “Human infection with respiratory adenovirus in China: a systematic review and meta-analysis”, authored by Mei-Chen Liu et al, described the epidemiological and clinical features of HAdV infections in China, from January 2009 to March 2021. In addition, the genetic and epidemiological characteristics of HAdVs were investigated.

I want to emphasize that the work is very well written and concisely covers all the epidemiological characteristics of HAdV respiratory infections.

The work carried out is a very important contribution to the knowledge of the classical and molecular epidemiology of HAdV in China and also provides valuable information worldwide.

Reviewer #3: The authors made changes to the manuscript to address the reviewers' comments. A few corrections are still necessary before publication, as pointed in the pdf file (attached).

PLOS authors have the option to publish the peer review history of their article (what does this mean?). If published, this will include your full peer review and any attached files.

Reviewer #1: No

Reviewer #2: No

Reviewer #3: No

Figure Files:

Data Requirements:

Reproducibility:

References

---

## [Editor Report · Decision Letter 2]

8 Feb 2023

Dear Dr Fang,

We are pleased to inform you that your manuscript 'Prevalence of human infection with respiratory adenovirus in China: a systematic review and meta-analysis' has been provisionally accepted for publication in PLOS Neglected Tropical Diseases.

Best regards,

Ali Rostami

Academic Editor

Elvina Viennet

Section Editor

---

## [Editor Report · Acceptance letter]

20 Feb 2023

Dear Professor Fang,

We are delighted to inform you that your manuscript, "Prevalence of human infection with respiratory adenovirus in China: a systematic review and meta-analysis," has been formally accepted for publication in PLOS Neglected Tropical Diseases.

Best regards,

Shaden Kamhawi

co-Editor-in-Chief

Paul Brindley

co-Editor-in-Chief
